# FROM DEMONSTRATIONS TO REWARDS: ALIGNMENT WITHOUT EXPLICIT HUMAN PREFERENCES

## ABSTRACT

One of the challenges of aligning large models with human preferences lies in both the data requirements and the technical complexities of current approaches. Predominant methods, such as RLHF, involve multiple steps, each demanding distinct types of data, including demonstrations data and preference data. In RLHF, human preferences are typically modeled through a reward model, which serves as a proxy to guide policy learning during the reinforcement learning stage, ultimately producing a policy aligned with human preferences. However, in this paper, we propose a fresh perspective on learning alignment based on inverse reinforcement learning principles, where the optimal policy is still derived from reward maximization. However, instead of relying on preference data, we directly learn the reward model from demonstration data. This new formulation offers the flexibility to be applied even when *only* demonstration data is available, a capability that current RLHF methods lack, and it also shows that demonstration data offers more utility than what conventional wisdom suggests. Our extensive evaluation, based on public reward benchmark and HuggingFace Open LLM Leaderboard, demonstrates that our approach compares favorably to state-of-the-art methods that rely solely on demonstration data.

## 1 INTRODUCTION

Despite the success of aligning methods to human preferences (OpenAI, 2023; Gemini-Team et al., 2023; Dubey et al., 2024; Gemini-Team, 2024), such as reinforcement learning from human feedback (RLHF) (Ziegler et al., 2019; Ouyang et al., 2022), these approaches are quite complex and require various types of data. For instance, RLHF involves multiple stages: the first being supervised fine-tuning (SFT), which uses human demonstration data consisting of input prompts and their corresponding human response pairs. Next is reward modeling, which relies on preference data, where input prompts

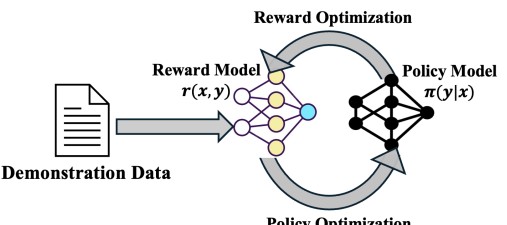

**LLM Alignment from Demonstration Data**

Figure 1: Illustration of the iterative RLHF pipeline for LLM alignment from demonstrations through inverse reinforcement learning .

are paired with multiple responses that are ranked based on relative preference (e.g., the preferred vs. non-preferred response). The final step is policy learning through reinforcement learning (RL), where only input prompts are required, and generated responses are scored using the reward model learned in the previous step. These technical and data complexities make training such models challenging and costly, particularly due to the need for diverse data types like human demonstrations and preference annotations, which require extensive human input for accurate labeling and ranking.

More specifically, one of the key challenges in RLHF is reward modeling, which acts as a proxy for human preferences. Existing approaches rely heavily on preference data to construct effective reward models demonstrating that a well-constructed reward function can boost model performance. (OpenAI, 2023; Gemini-Team et al., 2023; Dubey et al., 2024). Even direct preference alignment methods (Zhao et al., 2023; An et al., 2023; Rafailov et al., 2024; Ethayarajh et al., 2024) which avoid explicit reward modeling, still necessitate preference data to align the model with human preferences. Considering the complexity of collecting high-quality preference data—and setting aside the modeling challenges—the key question is: Can we extract preferences from demonstration data alone, given

that such data is known to contain valuable human preference information? One potential answer lies in the utilization of inverse reinforcement learning (IRL) to formulate this problem, which has the capability to learn both reward and policy simultaneously (Ng et al., 2000; Ziebart et al., 2008; Fu et al., 2017; Zeng et al., 2022b). This allows it to potentially learn preferences *directly* from demonstration data, even when relying solely on that data. Motivated by this observation, we propose a bi-level formulation based on IRL for the alignment problem, where the reward model and policy are modeled separately but learned in an interleaved manner. The interleaving between policy learning and reward learning helps to model preferences solely from demonstration data, although those preferences are implicit in the data (see Fig. 1). As our results show, this formulation indeed helps to improve the performance of the final model, particularly when compared to SFT, which also relies solely on demonstration data.

**Summary of contribution of this work:**

- We develop a new IRL-based method for alignment that relies solely on demonstration data, yet still improves the resulting model's performance. This method adopts a bi-level formulation, where the reward and policy are modeled separately. In this formulation, the policy is framed as the optimal solution to a KL-Divergence regularized policy optimization problem, constrained by a reward estimator. The reward model is optimized using the demonstration data to ensure that its corresponding optimal policy is the maximum likelihood estimator derived from the same dataset.

- We also present a theoretical connection with a recently proposed method SPIN (Chen et al., 2024), which can demonstrate that SPIN is a special variant derived from the IRL formulation from a bi-level optimization perspective.

- We demonstrate that the reward learned solely from demonstration data exhibits strong generative capabilities in assessing data quality. This is evidenced by evaluating the reward's accuracy on a hold-out preference dataset and a public reward benchmark (Lambert et al., 2024).

- Empirically, we extensively evaluate our proposed method by fine-tuning the 1B Pythia model on the TL;DR dataset (Stiennon et al., 2020; Huang et al., 2024) and the 7B Mistral model on the UltraChat dataset (Ding et al., 2023). Our numerical results demonstrate that the proposed method compares favorably to both SFT and SPIN, as indicated by higher win rates in evaluations conducted by ChatGPT and improved performance on downstream tasks in the Open LLM Leaderboard (Myrzakhan et al., 2024).

## 2 RELATED WORK

**Imitation learning** Imitation learning assumes the availability of a demonstration dataset containing expert data and focuses on learning from these demonstrations to match the expert's policy. Behavior cloning is one classic imitation learning algorithm which directly fits the demonstration data through supervised learning (Pomerleau, 1988). However, naively fitting sequential demonstrations can lead to distribution shift between demonstration trajectories and policy rollout. Moreover, since imitation learning where future transitions depend on previous actions violates the common i.i.d. assumptions made in statistical learning, naively fitting demonstrations trajectories can incur unfavorable regret bound which has quadratic dependence on the problem horizon (Ross & Bagnell, 2010; Ross et al., 2011). To address these challenges in imitation learning, it has been proposed to model the policy as the (optimal) solution to a MDP under a specific reward function (Osa et al., 2018). To learn a policy that effectively utilizes demonstration trajectories to match the expert policy, inverse reinforcement learning methods are proposed to search for one optimal reward estimator which can classify demonstration trajectories from other policy rollouts and guarantee that its corresponding optimal policy can imitate the observed expert behaviors in the demonstration dataset (Ziebart et al., 2008; Ho & Ermon, 2016; Fu et al., 2017; Zeng et al., 2022b; Garg et al., 2021).

**Imitation learning for language modeling** The connection between imitation and generative language modeling can be traced back to some adversarial training methods for text generation (Yu et al., 2017; Wu et al., 2021), although some of them may not draw explicit connection with imitation learning. There are also works of applying inverse RL methods to a specific text generation tasks, such as table-to-text generation (Ghosh et al., 2021), program generation (Ghosh & Srivastava, 2021), and summarization (Fu et al., 2022). To efficiently imitate the demonstration data, Chen et al. (2024) proposed an imitation learning algorithm, SPIN, which construct synthetic preference data through

pairing demonstration and model-generated continuations and then leverage direct preference method (Rafailov et al., 2024) for policy fine-tuning. Li et al. (2024) shows the connection between SPIN and IRL from a theoretical perspective and reveals that SPIN utilizes the parameterization technique developed in (Rafailov et al., 2024) to avoid explicit reward modelling and reward learning from computational simplicity. Although the parameterization technique introduced by Rafailov et al. (2024) allows skipping the reward modeling subroutine and reduces the complexities of RLHF, it faces challenges due to the distribution shift between preference data and model outputs. This shift can lead to instability during the training process (Xu et al., 2024; Ivison et al., 2024). It is wroth noting that while Li et al. (2024) discussed that the demonstration data can potentially benefit reward learning, it lacks practical algorithm for IRL methods which can iteratively enhance both reward model and policy by learning directly from demonstration data. More recently, Cundy & Ermon and Wulfmeier et al. (2024) apply the imitation learning algorithm IQLearn for the post-training step of large language models, making their approaches closer to ours. In contrast, we adopt the framework of maximum likelihood inverse RL (ML-IRL) and derive a different objective, with results on broader LLM post-training benchmarks. Similar with us, Sun (2024) draws a connection between inverse RL and the supervised fine-tuning problem of LLMs, but lacks both algorithmic implementation and experimental studies.

## 3 PRELIMINARIES

We formulate the alignment problem of auto-regressive language models as an Markov decision process (MDP) following Ouyang et al. (2022). For a language model $\pi$, we denote its probability of generating a completion as $\pi(\boldsymbol{y}|\boldsymbol{x})$, where $\boldsymbol{x} = [x_1, x_2, \cdots, x_n]$ denotes the sequence of tokens in the input prompt and $\boldsymbol{y} = [y_1, y_2, \cdots, y_H]$ denotes the sequence of tokens in the model generated continuation. The language model generates each token auto-regressively in a sequential manner as $\pi(\boldsymbol{y}|\boldsymbol{x}) = \prod_{h=1}^{H} \pi(y_h|\boldsymbol{x}, \boldsymbol{y}_{1:h-1})$, where each step $h$ is viewed as a time-step in the MDP.

The current predominant method for aligning models with human preferences is RLHF (Ouyang et al., 2022), which comprises mutiple stages, as outlined earlier. In the first stage, SFT, a high-quality human demonstration dataset $\mathcal{D} = \{(\boldsymbol{x}^i, \boldsymbol{y}^i)\}_{i=1}^{N}$ is used to fine-tune the pre-trained model using the following maximum-likelihood objective:

$$\min_{\phi} \ell_{\text{SFT}}(\phi) := -\mathbb{E}_{(\boldsymbol{x},\boldsymbol{y})\sim\mathcal{D}} \left[\log \pi\left(\boldsymbol{y}|\boldsymbol{x}; \phi\right)\right]. \tag{1}$$

In the reinforcement learning literature, this method is also known as behavior cloning (Osa et al., 2018). Notably, theoretical analyses of behavior cloning (Ross & Bagnell, 2010; Ross et al., 2011) indicate that directly fitting sequential demonstration data in a MDP can result in unfavorable regret bounds, exhibiting quadratic dependence on the problem horizon. These insights suggest that SFT, as a form of behavior cloning, may not be the most effective approach for learning from demonstration in MDPs. To improve policy learning from sequential demonstrations, methods such as imitation learning and IRL (Ng et al., 2000; Ho & Ermon, 2016; Osa et al., 2018; Zeng et al., 2022b) have been proposed, offering superior performance compared to naive behavior cloning methods.

In the second stage, there are wo prominent classes of RLHF algorithms. One is explicitly building a parametric reward model and then fine-tuning the policy with online RL methods and the other is directly learning a policy from preference data. We refer to them as reward-based and reward-free methods in this paper. Reward-based RLHF approaches first train a reward model $r(\boldsymbol{x}, \boldsymbol{y}; \theta)$ by separating the score between preferred completion and non-preferred completion in a preference dataset $\mathcal{D}_{\mathcal{P}} := \{(\boldsymbol{x}, \boldsymbol{y}_w, \boldsymbol{y}_l)\}$ where $\boldsymbol{y}_w$ is preferred one over $\boldsymbol{y}_l$ according to the annotation from human annotator. (see e.g., Christiano et al. (2017); Stiennon et al. (2020); Ouyang et al. (2022)). More specifically, RLHF methods follow the Bradley-Terry model (Bradley & Terry, 1952) which assumes that the distribution of preference label under one reward model $r(\boldsymbol{x}, \boldsymbol{y})$ is represented as $\mathbb{P}(\boldsymbol{y}_w \succ \boldsymbol{y}_l \mid \boldsymbol{x}) = \sigma(r(\boldsymbol{x}, \boldsymbol{y}_w) - r(\boldsymbol{x}, \boldsymbol{y}_l))$, where $\sigma(\cdot)$ is the sigmoid function. Therefore one can derive the reward learning objective from the maximum log-likelihood (MLE) on Bradley-Terry model:

$$\max_{\theta} \ell_{\text{RM}}(\theta) := \mathbb{E}_{(\boldsymbol{x},\boldsymbol{y}_w,\boldsymbol{y}_l)\sim\mathcal{D}_{\mathcal{P}}} \left[\log\left(\sigma\left(r(\boldsymbol{x}, \boldsymbol{y}_w; \theta) - r(\boldsymbol{x}, \boldsymbol{y}_l; \theta)\right)\right)\right]. \tag{2}$$

After learned the reward model, various of online RL approaches can be used to fine-tune the policy from its own generation, for examples proximal policy optimization (PPO) (Schulman et al., 2017),

variants of REINFORCE (Ahmadian et al., 2024; Li et al., 2023) and reward-ranked fine-tuning (Dong et al., 2023). The most commonly used objective in this stage is KL-regularized reward maximization:

$$\max_{\pi} \; \ell_{\mathrm{RL}}(\pi) := \mathbb{E}_{\boldsymbol{x} \sim \mu, \boldsymbol{y} \sim \pi(\cdot|\boldsymbol{x})} \left[ r(\boldsymbol{x}, \boldsymbol{y}; \theta) \right] - \beta \mathbb{E}_{\boldsymbol{x} \sim \mu}[D_{\mathrm{KL}}(\pi\left(\cdot|\boldsymbol{x}\right) \| \pi_{\mathrm{ref}}\left(\cdot|\boldsymbol{x}\right))], \tag{3}$$

where $\pi_{\mathrm{ref}}$ is a fixed reference model (usually the SFT model) and $\mu(\cdot)$ denotes the prompt distribution over one prompt dataset. Here, in the policy optimization problem, the regularization of the KL divergence between the policy model $\pi$ and reference model $\pi_{\mathrm{ref}}$ ensures that the langugae model will not deviate from the reference model too much. The advantage of RLHF over SFT observed by Stiennon et al. (2020); Ouyang et al. (2022); Gemini-Team et al. (2023) comes from two aspects hypothetically: the generalization ability from reward model and learning on self-generated sequence.

Reward-free RLHF approaches (Rafailov et al., 2024; An et al., 2023) is an alternative to the classical reward-based RLHF by shortcutting the reward learning step, or implicitly learning it together with the policy learning. As an example, Direct Preference Optimization (DPO) (Rafailov et al., 2024) propose to incorporate reward learning implicitly by utilizing the structure of the optimal solution of the RL problem in Eq. (3). Based on that, DPO derives its objective as below:

$$\max_{\pi} \; \mathbb{E}_{(\boldsymbol{x}, \boldsymbol{y}_w, \boldsymbol{y}_l) \sim \mathcal{D}_{\mathcal{P}}} \left[ \log \left( \sigma \big( \beta \log \left( \frac{\pi(\boldsymbol{y}_w|\boldsymbol{x})}{\pi_{\mathrm{ref}}(\boldsymbol{y}_w|\boldsymbol{x})} \right) - \beta \log \left( \frac{\pi(\boldsymbol{y}_l|\boldsymbol{x})}{\pi_{\mathrm{ref}}(\boldsymbol{y}_l|\boldsymbol{x})} \right) \big) \right) \right]. \tag{4}$$

## 4 PROBLEM FORMULATION

As we have mentioned in Sec. 1 and also motivated by the theoretical understanding developed in (Ross & Bagnell, 2010; Ross et al., 2011), SFT or equivalently behavior cloning can incur unfavorable error bound which has quadratic dependence on the problem horizon when learning from demonstration data with sequential structure. To bridge the gap between the current SFT method and imitation learning methods in RL literature (Ng et al., 2000; Ho & Ermon, 2016; Osa et al., 2018; Zeng et al., 2022b), we consider a maximum likelihood formulation for IRL which this approach allows for the learning of a reward model and fine-tunes the SFT model using demonstration data.

### 4.1 A MAXIMUM LIKELIHOOD FORMULATION FOR REWARD LEARNING AND POLICY FINE-TUNING

Given a demonstration dataset $\mathcal{D}$, the challenge lies in learning a reward model that aligns its corresponding policy with the demonstration data and effectively captures the implicit human preferences contained within it. Unlike the standard RLHF, where the reward model is trained on a dataset of pairwise comparisons that explicitly represents human preferences, learning rewards from a demonstration dataset presents additional challenges due to the lack of explicit preferences in this data. Instead, this dataset contains only implicit preferences. Motivated by inverse reinforcement learning (IRL) based approaches (Ziebart et al., 2008; 2013; Fu et al., 2017; Zeng et al., 2022a;b), we propose an IRL formulation grounded in maximum likelihood estimation to align the model with the demonstration dataset. Our maximum likelihood formulation aims to learn an "optimal" reward model such that its corresponding policy serves as the maximum likelihood estimator over the demonstration dataset. Here, we present our proposed formulation as follows:

$$\max_{\theta} \quad L(\theta) := \mathbb{E}_{\boldsymbol{x} \sim \mu(\cdot), \boldsymbol{y} \sim \pi^{\mathrm{E}}(\cdot|\boldsymbol{x})} \left[ \log \pi_{r_\theta}^*(\boldsymbol{y}|\boldsymbol{x}) \right] \tag{5a}$$

$$s.t \quad \pi_{r_\theta}^* := \arg\max_{\pi} \; \mathbb{E}_{\boldsymbol{x} \sim \mu(\cdot), \boldsymbol{y} \sim \pi(\cdot|\boldsymbol{x})} \left[ r(\boldsymbol{x}, \boldsymbol{y}; \theta) - D_{\mathrm{KL}}\big(\pi(\cdot|\boldsymbol{x}) \| \pi_{\mathrm{ref}}(\cdot|\boldsymbol{x})\big) \right], \tag{5b}$$

where $\mu(\cdot)$ denotes the distribution of the prompt, $\pi$ denotes a policy model which generates continuations from given prompts, $\pi^{\mathrm{E}}$ denotes the expert-level policy which can generates high-quality demonstration continuations and $\pi_{\mathrm{ref}}$ denotes one reference model which is usually chosen as the SFT model in LLM alignment. Moreover, $r(s, a; \theta)$ is the parameterized reward model and $\pi_{r_\theta}^*$ denotes the optimal policy to a KL Divergence regularized policy optimization problem when the reward model is $r(\cdot, \cdot; \theta)$.

We now make some remarks about our maximum likelihood formulation of our alignment algorithm. First, the problem takes the form of a *bi-level* optimization problem, where the *upper-level* problem

Eq. (5a) optimizes the reward parameter $\theta$, while the *lower-level* problem Eq. (5b) describes the corresponding policy $\pi_{r_\theta}^*$ as the solution to an KL Divergence regularized policy optimization problem (Stiennon et al., 2020; Ouyang et al., 2022). As a remark, despite that our maximum likelihood formulation Eq. (5) establishes one framework to imitate the expert policy and estimate the reward model from demonstration dataset, it is impractical to continuously sample demonstration generations from the expert policy $\pi^E$ in Eq. (5a) since the expert policy is unknown and only one observed demonstration dataset is available. To resolve this issue, we instead replace the expert policy by one fixed demonstration dataset which contains finite samples.

Given a demonstration dataset $\mathcal{D} := \{(\boldsymbol{x}, \boldsymbol{y})\}$, we propose a surrogate objective $\widehat{L}(\theta; \mathcal{D})$ which approximates the maximum likelihood formulation Eq. (5) with finite demonstration data. Here, we consider the following surrogate problem:

$$\max_\theta \widehat{L}(\theta; \mathcal{D}) := \mathbb{E}_{(\boldsymbol{x}, \boldsymbol{y}) \sim \mathcal{D}} \big[ r(\boldsymbol{x}, \boldsymbol{y}; \theta) + \log \pi_{\text{ref}}(\boldsymbol{y}|\boldsymbol{x}) \big]$$

$$- \mathbb{E}_{\boldsymbol{x} \sim \mu(\cdot), \boldsymbol{y} \sim \pi_{r_\theta}^*(\cdot|\boldsymbol{x})} \left[ r(\boldsymbol{x}, \boldsymbol{y}; \theta) - D_{\text{KL}} \left( \pi_{r_\theta}^*(\cdot|\boldsymbol{x}) \| \pi_{\text{ref}}(\cdot|\boldsymbol{x}) \right) \right] \tag{6}$$

where the policy $\pi_{r_\theta}^*$ denotes the optimal policy corresponding to the policy optimization problem defined in Eq. (5b) when the reward model is parameterized by the parameter $\theta$.

Based on the surrogate estimation problem Eq. (6), we show that the IRL for LLM alignment problem defined in Eq. (5) can be accurately approximated with a *finite* set of high-quality generations when the "expert-level" generative model (or data source) $\pi^E$ is not known. In particular, below we show under a mild assumption about the boundedness of the reward score and the reference model, $\widehat{L}(\theta; \mathcal{D})$ can well-approximate $L(\theta)$ when the offline demonstration dataset includes sufficient number of high-quality generations.

**Assumption 4.1.** *For any reward parameter $\theta$, the following condition holds:*

$$0 \le r(\boldsymbol{x}, \boldsymbol{y}; \theta) \le C_r, \quad C_p \le \log \pi_{\text{ref}}(\boldsymbol{y}|\boldsymbol{x}) < 0 \quad \forall \boldsymbol{x}, \boldsymbol{y} \tag{7}$$

*where $C_r > 0$ and $C_p < 0$ are fixed constants.*

**Lemma 4.1.** *Suppose Assumption 4.1 hold. Consider the likelihood function $L(\theta)$ in Eq. (5a) and its surrogate empirical version $\widehat{L}(\theta; \mathcal{D})$ defined in Eq. (6). Then, with probability greater than $1 - \delta$, we have:*

$$|L(\theta) - \widehat{L}(\theta; \mathcal{D})| \le (C_r - C_p) \sqrt{\frac{\ln(2/\delta)}{2|\mathcal{D}|}}. \tag{8}$$

The proof of Lemma 4.1 can be found in Appendix.

## 5 THE PROPOSED ALGORITHM

We are now ready to design algorithms for the proposed maximum likelihood formulation Eq. (5) aimed at aligning large language models (LLMs) using demonstration data. To begin with, first note that the maximum likelihood formulation 5 and its surrogate estimation problem Eq. (6) takes a hierarchical form, and it belongs to the class of problem named *bi-level* optimization. Generally speaking, bi-level problems are not easy to optimize since they have a nested structure for two optimization problems. In this section, we will propose one computationally tractable algorithm for both reward learning and policy fine-tuning to solve the LLM alignment problem 6.

Before presenting the details, we emphasize that throughout this section, we aim to explicitly identify *both* an optimal policy $\pi_{r_\theta}^*$ and a corresponding reward estimate $r(\cdot, \cdot; \theta)$ that align with the demonstration data. Specifically, the policy $\pi_{r_\theta}^*$ is considered an optimal solution with respect to the reward estimate $r(\cdot, \cdot; \theta)$, as defined by the policy optimization problem in equation Eq. (5b). Given this optimal policy constraint relative to a specific reward estimate, we propose an algorithm to tackle this single-stage, bi-level problem. It is important to note that our approach is a departure from popular methods like DPO (Rafailov et al., 2024), which directly optimize a fixed loss function (see Eq. (4)) without explicitly modeling the reward.

---

**Algorithm 1:** *Joint Reward Learning and Policy Fine-tuning from Demonstrations*

**Input:** A demonstration dataset $\mathcal{D}$, a reference model $\pi_{\text{ref}}$ and $K$ the number of iterations.
**for** $k = 0, \ldots, K - 1$ **do**
    **Policy Alignment:** Run policy optimization subroutine (like PPO) to update the policy:

$$\pi_{k+1}(\boldsymbol{y}|\boldsymbol{x}) \propto \pi_{\text{ref}}(\boldsymbol{y}|\boldsymbol{x}) \exp(r(\boldsymbol{x}, \boldsymbol{y}; \theta_k))$$

    **Reward Alignment:** Construct synthetic preference data and optimizing the following problem:

$$\theta_{k+1} := \arg\min_\theta -\mathbb{E}_{(\boldsymbol{x},\boldsymbol{y}) \sim \mathcal{D}, \boldsymbol{y}' \sim \pi_{k+1}(\cdot|\boldsymbol{x})} \Big[ \log \Big( \sigma \big( r(\boldsymbol{x}, \boldsymbol{y}; \theta) - r(\boldsymbol{x}, \boldsymbol{y}'; \theta) \big) \Big) \Big]$$

**end for**
**Output:** Estimated Reward Model $r(\cdot, \cdot; \theta_K)$ and Policy Model $\pi_K(\cdot|\cdot)$

---

Our algorithm operates by alternating between two key steps: 1) **Policy Alignment Step**, where we perform a policy improvement step towards solving Eq. (5b) under a fixed reward function $r(\cdot, \cdot; \theta)$, effectively aligning the policy with the current reward estimate, 2) **Reward Alignment Step**, where we update the reward parameters $\theta$ using a stochastic gradient estimator to align the reward model with the demonstration dataset. This training loop allows the policy and reward model to be fine-tuned iteratively, potentially leading to better alignment with the demonstration data compared to standard SFT methods which treat the loss function Eq. (1) as fixed.

In the following sections, we delve into each of these steps in greater detailw, providing theoretical insights and practical implementation considerations.

**Policy Alignment Step.** From our earlier discussion, we know that the optimal policy $\pi_{r_\theta}^*$ corresponds to the optimal solution to the policy optimization problem Eq. (5b) under a fixed reward model $r(\cdot, \cdot; \theta)$. To tackle such policy optimization problem, one can adopt the standard approaches such as the well-known proximal policy optimization (PPO) (Schulman et al., 2017) algorithm to obtain an approximate optimal policy. As a remark, due to some practical difficulties for implementing PPO to fine-tune LLMs (like heavy memory cost and laborious hyper-parameter tuning), it is possible to consider a simpler method than running PPO to obtain the optimal policy. Some of recently proposed RLHF methods like REINFORCE-type variants (Li et al., 2023; Ahmadian et al., 2024) and reward ranked fine-tuning (Dong et al., 2023) provides more computationally tractable alternatives to PPO for fine-tuning LLMs under one estimated reward model. It is important to note that, the point of the above discussion is that all of these different choices for policy optimization methods can be incorporated into our policy alignment step.

From a theoretical perspective, based on (Cen et al., 2022; Zeng et al., 2022b; Ji et al., 2024), one can perform a "soft policy iteration" to obtain one updated policy estimator $\pi_{k+1}$ in each iteration $k$ as below:

$$\pi_{k+1}(\boldsymbol{y}|\boldsymbol{x}) \propto \pi_{\text{ref}}(\boldsymbol{y}|\boldsymbol{x}) \exp\big(r(\boldsymbol{x}, \boldsymbol{y}; \theta)\big), \quad \forall s \in \mathcal{S}, a \in \mathcal{A}. \tag{9}$$

To approximate the closed-form optimal policy in practice, one can utilize the popular policy optimization pipeline for fine-tuning LLMs under one reward model (Schulman et al., 2017; Ahmadian et al., 2024).

**Reward Optimization Step.** We propose to use a stochastic gradient-type algorithm to optimize $\theta$. Towards this end, let us first derive the exact gradient $\nabla\widehat{L}(\theta; \mathcal{D})$. See Appendix for detailed proof.

**Lemma 5.1.** *The gradient of the finite-sample surrogate objective function $\nabla\widehat{L}(\theta; \mathcal{D})$ can be expressed as follows:*

$$\nabla\widehat{L}(\theta; \mathcal{D}) = \mathbb{E}_{(\boldsymbol{x},\boldsymbol{y}) \sim \mathcal{D}}\big[\nabla_\theta r(\boldsymbol{x}, \boldsymbol{y}; \theta)\big] - \mathbb{E}_{\boldsymbol{x} \sim \mu(\cdot), \boldsymbol{y} \sim \pi_{r_\theta}^*(\cdot|\boldsymbol{x})}\big[\nabla_\theta r(\boldsymbol{x}, \boldsymbol{y}; \theta)\big]. \tag{10}$$

To obtain stochastic estimators of the exact gradient $\nabla\widehat{L}(\theta; \mathcal{D})$, we take two approximation steps: 1) approximate the optimal policy $\pi_{k+1}$ in Eq. (9) through running a finite policy optimization steps in the RL subroutine since repeatedly estimating the optimal policy under each reward estimator can lead to computational burden; 2) sample one batch of demonstration data from the demonstration dataset $\mathcal{D}$; 3) sample model-generated data from the current policy estimator. Theoretically, as long

as the policy estimator achieves policy improvement in each IRL iterations, the training pipeline can be stable in such approximation and converge to the optimal solution (Zeng et al., 2022a).

Intuitively, following the reward gradient expression as shown in Eq. (10), if the model-generated data from the current policy $\pi_{k+1}$ has not matched the demonstration dataset $\mathcal{D}$ yet, then the reward score should be improved by going towards the direction suggested by the demonstration data, while *going away* from those generated by the current policy. Similar to the BTL model, from the gradient expression Eq. (2), it is clear that the algorithm will find the reward update direction that increases the gap between the reward of the real samples (demonstrations) and the synthetic ones (model generated continuations). Hence, as for each reward optimization step at iteration $k$, one can construct the following loss function to update the reward parameter $\theta$ through constructing synthetic preference data through pairing the demonstration data with the model generations:

$$\min_{\theta} L_{\mathrm{RM}}(\theta; \mathcal{D}) := -\mathbb{E}_{(\boldsymbol{x}, \boldsymbol{y}) \sim \mathcal{D}, \boldsymbol{y}' \sim \pi_{k+1}(\cdot | \boldsymbol{x})} \Big[ \log \Big( \sigma \big( r(\boldsymbol{x}, \boldsymbol{y}; \theta) - r(\boldsymbol{x}, \boldsymbol{y}'; \theta) \big) \Big) \Big]. \quad (11)$$

In summary, the proposed algorithm for solving Eq. (6) is given in Alg. 1.

## 6 EXPERIMENTAL RESULTS

In this section, we evaluate the effectiveness of our proposed method through comprehensive experiments on two distinct datasets: the TL;DR dataset (Stiennon et al., 2020) for summarization tasks and the UltraChat dataset (Ding et al., 2023) for dialogue generation. Our aim is to demonstrate that high-quality demonstration data can be leveraged to construct synthetic preference datasets, which in turn can significantly improve both reward models and policy models without the sole reliance on human-annotated preferences. The experimental results show that our approach not only enhances model performance in terms of alignment with human judgments but also achieves reward learning without explicit human preferences.

### 6.1 EXPERIMENTS ON THE TL;DR DATASET

In this experiment, we aim to train a language model for text summarization tasks using the TL;DR dataset (Stiennon et al., 2020), available on Hugging Face. We use all prompts from the TL;DR dataset as our prompt dataset. To create a demonstration dataset, we generate 10,000 high-quality summaries using a 6.9B parameter Pythia checkpoint (Huang et al., 2024) that was trained via a RLHF pipeline with human-annotated preference data. This model is publicly available on Hugging Face[1].

In our IRL pipeline which iteratively updating from policy and reward model through utilizing the demonstration data, we begin by performing supervised fine-tuning (SFT) on a pretrained 1B parameter Pythia model using the generated demonstration dataset, resulting in our initial SFT model. At each iteration, we construct a preference dataset by labeling the summaries generated from the 6.9B PPO-trained checkpoint as preferred and the outputs generated by our current 1B Pythia model as non-preferred. Using this preference dataset, we train a reward model initialized from our 1B SFT model. We then apply the Proximal Policy Optimization (PPO) algorithm, guided by the estimated reward model, to further fine-tune our 1B Pythia model, enhancing its performance beyond the initial SFT checkpoint. For the PPO algorithm setup, we follow the hyperparameters and experimental pipeline detailed in (Huang et al., 2024). Our implementation is consistent with their codebase[2].

We present our numerical results in Figure 2, showcasing the performance of the proposed iterative RLHF pipeline from three perspectives: (1) reward model accuracy, (2) reward scores measured by a 6.9B ground-truth reward model (Huang et al., 2024) trained on the human-annotated TL;DR preference dataset, and (3) generating continuations from the prompt in the test dataset of the TL;DR dataset and then evaluate the win rates by GPT-4o to compare the text summarizations generated by IRL policy models and the 6.9B PPO-trained checkpoint.

Figure 2 (a) presents the accuracy of our estimated 1B reward model on a human-annotated preference dataset of TL;DR [3], which serves as a hold-out, out-of-distribution dataset since it is not used during

---

[1] https://huggingface.co/vwxyzjn

[2] https://github.com/vwxyzjn/summarize_from_feedback_details

[3] https://huggingface.co/datasets/openai/summarize_from_feedback

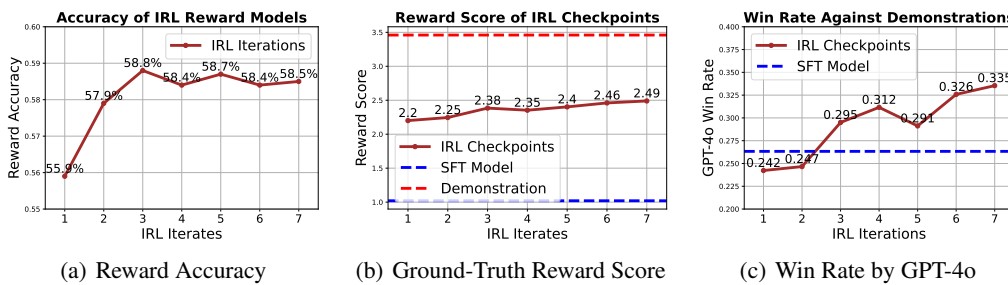

| (a) Reward Accuracy | (b) Ground-Truth Reward Score | (c) Win Rate by GPT-4o |

Figure 2: Numerical Results of IRL Iterations with high-quality SFT data.

our reward learning process. As shown in the figure, our IRL algorithm improves the reward model's accuracy over successive iterations, indicating better alignment with human preferences. Figure 2 (b) illustrates the ground-truth reward scores assigned by the 6.9B Pythia reward model to summaries generated by our 1B policy model. The results indicate that our iterative RLHF method enhances the model's performance over iterations, as reflected by increasing reward scores. Additionally, Figure 2 (c) presents the win rate of our 1B policy model compared to the 6.9B PPO-trained checkpoint, as evaluated by GPT-4. The iterative RLHF pipeline increases the SFT model's win rate from 24% to 33%, signifying that the proposed IRL method significantly outperforms the original SFT pipeline when learning from high-quality demonstrations.

These findings collectively suggest that our approach effectively leverages high-quality demonstration data to construct meaningful preference datasets, leading to improvements in both the reward model and the policy model. The IRL pipeline not only enhances the alignment with human judgments but also achieves performance gains that are notable given the smaller size of the 1B parameter model compared to the 6.9B parameter baseline. By following this methodology, we demonstrate that even smaller models can achieve significant performance improvements through iterative RLHF processes towards imitating one larger language model when high-quality demonstrations are available.

## 6.2 EXPERIMENTS ON THE ULTRACHAT DATASET

In this section, we present experiments demonstrating our proposed method applied to the UltraChat dataset (Ding et al., 2023), a high-quality dialogue dataset. For our experiments, we initialize both policy model and reward model from the checkpoint `alignment-handbook/zephyr-7b-sft-full`[4].

Since UltraChat is a supervised fine-tuning (SFT) dataset containing only demonstration data, we construct a synthetic preference dataset to train our reward model. Specifically, in the reward learning step for each IRL iteration, we treat the demonstration data from UltraChat as the preferred responses and the outputs generated by the IRL policy model as the rejected responses. This approach allows us to create preference pairs without requiring explicit human annotations.

We evaluate our estimated reward models using the `allenai/reward-bench`(Lambert et al., 2024), assessing performance across various categories relevant to language understanding and generation. The results, illustrated in Figure 3, show that the reward model trained through the proposed IRL method achieves significant improvements compared to both the base model (initialized from the SFT model) and the implicit reward model extracted from the policy model trained using SPIN (Chen et al., 2024). These findings indicate that high-quality demonstration datasets can effectively enhance reward models through leveraging IRL method which can construct synthetic preference pairs through pairing high-quality demonstrations and model generations.

In each iteration of our IRL process, after updating the reward model, we fine-tune the previous IRL policy checkpoints using policy optimization methods guided by the estimated reward model. For the policy optimization subroutine, we follow the implementation details provided in the codebase of

---

[4]https://huggingface.co/alignment-handbook/zephyr-7b-sft-full

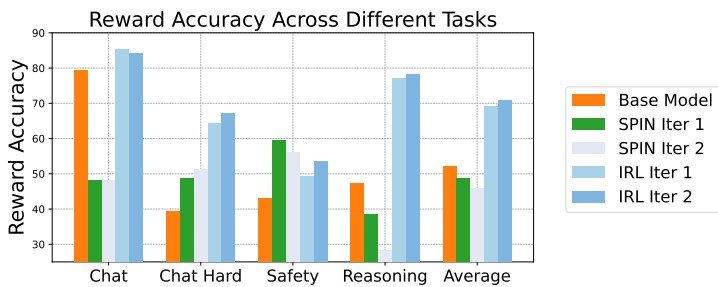

Figure 3: Evaluation of Reward Models on Reward Bench

(Dong et al., 2024)[5]. We employ the Best-of-N sampling strategy as our policy trainer, which offers a memory-efficient approach for training large language models with limited computation resources.

To evaluate the effectiveness of our approach, we assessed our fine-tuned models on the Open LLM Leaderboard (Gao et al., 2023). As shown in Table 2, our method outperforms both the zephyr-7b-sft-full baseline and the SPIN method in Iterations 1 and 2, providing further evidence of the applicability and effectiveness of our approach. These results highlight the potential of leveraging high-quality demonstrations and synthetic preferences to enhance language model performance in dialogue generation tasks.

By utilizing synthetic preference data derived from high-quality demonstrations, our approach effectively strengthens the reward model, which in turn enhances the policy model through iterative training. This strategy reduces the reliance on costly human-annotated preference data and demonstrates a scalable method for improving LLMs with high-quality demonstration dataset.

| Reward Model | Chat | Chat Hard | Safety | Reasoning | Average |
|---|---|---|---|---|---|
| Base (zephyr-7b-sft-full) | 79.19% | 39.25% | 42.94% | 47.29% | 52.16% |
| SPIN Iter 1 | 48.04% | 48.79% | **59.5%** | 38.59% | 48.73% |
| SPIN Iter 2 | 48.05% | 51.32% | 56.0% | 28.14% | 45.88% |
| IRL Iter 1 | **85.20%** | 64.25% | 49.34% | 77.14% | 68.98% |
| IRL Iter 2 | 84.22% | **67.0%** | 53.53% | **78.1%** | **70.71%** |

Table 1: Performance of Reward Models in Reward-Bench.

| Tasks | Arc Challenge | TruthfulQA MC2 | Winogrande | GSM8k | HellaSwag | MMLU | Avg |
|---|---|---|---|---|---|---|---|
| Metrics | acc_norm | acc | acc | strict-match | acc_norm | acc | |
| zephyr-7b-sft-full | 57.34% | 40.37% | 76.24% | 32.30% | 81.08% | 58.85% | 57.70% |
| SPIN Iter 1 | 60.41% | 41.15% | 76.80% | **33.89%** | 82.94% | 58.78% | 59.00% |
| SPIN Iter 2 | 60.41% | 40.94% | 75.85% | 32.98% | **83.01%** | **59.04%** | 58.71% |
| IRL Iter 1 | 59.73% | 45.32% | **77.03%** | 33.81% | 82.26% | 58.36% | 59.42% |
| IRL Iter 2 | **60.41%** | **46.30%** | 76.24% | 33.66% | 82.33% | 57.98% | **59.49%** |

Table 2: Performance comparison between IRL and SPIN across the six benchmark datasets.

## 7 CONCLUSION

In this paper, we propose a new formulation for the alignment problem based on the IRL framework that utilizes only demonstration data. This approach enables us to simultaneously learn both a reward model and a policy model, resulting in a method that is more efficient than other demonstration-only methods, such as SFT. Our extensive experiments, on public reward benchmark and the Hugging Face Open LLM Leaderboard, demonstrate performance improvements over existing SFT-based methods solely on demonstration data. These findings underscore that demonstration data offers greater utility than conventional wisdom suggests.

---

[5]https://github.com/RLHFlow/Online-RLHF

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

APPENDIX

# A  EXPERIMENT DETAILS

In this section, we include the details in hyperparameters for our experiment in TL;DR and UltraChat. It is worth mentioning that we conducted a minimal random hyper-parameters search for the experiments in this paper and we mostly follow standard and readily available settings for the experiments whenever it is applicable.

## A.1  EXPERIMENT ON TL;DR FOR TRAINING PYTHIA-1B

In the experiment of TL;DR, we use the TL;DR dataset to train the model *pythia-1* For the RL trainer we used in our IRL pipeline, we utilize the PPO trainer. Here, we include the hyper-parameters for both reward modeling and PPO trainer as below.

Table 3: Reward modeling hyperparameters

| Hyperparameter | Default Value |
|---|---|
| Number of Train Epochs | 1 |
| Optimizer | AdamW ($\epsilon = 1e^{-5}$, lr $= 3e^{-6}$) |
| Scheduler | Cosine |
| Batch Size | 64 |

Table 4: PPO hyperparameters

| Hyperparameter | Default Value |
|---|---|
| Optimizer | AdamW ($\epsilon = 1e^{-5}$, lr $= 3e^{-6}$) |
| Scheduler | Linear |
| Batch Size | 512 |
| $\beta$ (KL Penalty Coefficient for RLHF) | 0.05 |
| $\gamma$ (Discount Factor) | 1.0 |
| $\lambda$ (for GAE) | 0.95 |
| $N_{\text{mb}}$ Number of Mini-batches | 1 |
| $K$ (Number of PPO Update Iterations Per Epoch) | 4 |
| $\epsilon$ (PPO's Policy Clipping Coefficient) | 0.2 |
| $\hat{\epsilon}$ (Value Clipping Coefficient) | 0.2 |
| $c_1$ (Value Function Coefficient) | 0.1 |
| Value Function Loss Clipping | True |
| Sampling Temperature | 0.7 |

## A.2  EXPERIMENT ON ULTRACHAT FOR TRAINING ZEPHYR-7B-SFT-FULL

In the experiment of utilizing the UltraChat dataset to finetune *zephyr-7b-sft-full*, we utilize the Best-of-N algorithm as the RL trainer in our IRL pipeline. Here, we include the hyper-parameter details as below:

Table 5: Reward modeling hyperparameters

| Hyperparameter | Default Value |
|---|---|
| Number of Train Epochs | 1 |
| Optimizer | AdamW ($\epsilon = 1e^{-5}$, lr $= 5e^{-6}$) |
| Scheduler | Cosine |
| Batch Size | 64 |

Table 6: Best-of-N hyperparameters

| Hyperparameter | Default Value |
| --- | --- |
| Optimizer | AdamW ($\epsilon = 1e^{-5}$, lr $= 5e^{-7}$) |
| Scheduler | Cosine |
| Batch Size | 64 |
| KL coefficient | 0.1 |
| Best-of-n | 32 |

# B    PROOF OF LEMMA 4.1

*Proof.* For the policy optimization problem defined in Eq. (5b), one can show that $\pi_{r_\theta}^*$ is the following closed-form expression:

$$\pi_{r_\theta}^*(\boldsymbol{y}|\boldsymbol{x}) = \frac{\pi_{\text{ref}}(\boldsymbol{y}|\boldsymbol{x}) \exp\left(r(\boldsymbol{x}, \boldsymbol{y}; \theta)\right)}{\sum_{\tilde{y}} \pi_{\text{ref}}(\tilde{\boldsymbol{y}}|\boldsymbol{x}) \exp\left(r(\boldsymbol{x}, \tilde{\boldsymbol{y}}; \theta)\right)}, \quad \forall \boldsymbol{x}, \boldsymbol{y}. \tag{12}$$

Then we can re-write the likelihood objective $L(\theta)$ defined in Eq. (5) as below:

$$L(\theta) = \mathbb{E}_{\boldsymbol{x} \sim \mu(\cdot), \boldsymbol{y} \sim \pi^{\text{E}}(\cdot|\boldsymbol{x})} \left[ \log \pi_{r_\theta}^*(\boldsymbol{y}|\boldsymbol{x}) \right]$$

$$= \mathbb{E}_{\boldsymbol{x} \sim \mu(\cdot), \boldsymbol{y} \sim \pi^{\text{E}}(\cdot|\boldsymbol{x})} \left[ \log \left( \frac{\pi_{\text{ref}}(\boldsymbol{y}|\boldsymbol{x}) \exp\left(r(\boldsymbol{x}, \boldsymbol{y}; \theta)\right)}{\sum_{\tilde{y}} \pi_{\text{ref}}(\tilde{\boldsymbol{y}}|\boldsymbol{x}) \exp\left(r(\boldsymbol{x}, \tilde{\boldsymbol{y}}; \theta)\right)} \right) \right]$$

$$= \mathbb{E}_{\boldsymbol{x} \sim \mu(\cdot), \boldsymbol{y} \sim \pi^{\text{E}}(\cdot|\boldsymbol{x})} \left[ \log\left(\pi_{\text{ref}}(\boldsymbol{y}|\boldsymbol{x}) \exp\left(r(\boldsymbol{x}, \boldsymbol{y}; \theta)\right)\right) - \log\left(\sum_{\tilde{y}} \pi_{\text{ref}}(\tilde{\boldsymbol{y}}|\boldsymbol{x}) \exp\left(r(\boldsymbol{x}, \tilde{\boldsymbol{y}}; \theta)\right)\right) \right]$$

$$= \mathbb{E}_{\boldsymbol{x} \sim \mu(\cdot), \boldsymbol{y} \sim \pi^{\text{E}}(\cdot|\boldsymbol{x})} \left[ r(\boldsymbol{x}, \boldsymbol{y}; \theta) + \log \pi_{\text{ref}}(\boldsymbol{y}|\boldsymbol{x}) \right] - \mathbb{E}_{\boldsymbol{x} \sim \mu(\cdot), \boldsymbol{y} \sim \pi_{r_\theta}^*(\cdot|\boldsymbol{x})} \left[ r(\boldsymbol{x}, \boldsymbol{y}; \theta) - D_{\text{KL}}\left(\pi_{r_\theta}^*(\cdot|\boldsymbol{x}) \| \pi_{\text{ref}}(\cdot|\boldsymbol{x})\right) \right].$$

Moreover, given a dataset of collected expert trajectories, we have defined the estimation problem $\widehat{L}(\theta; \mathcal{D})$ as below:

$$\widehat{L}(\theta; \mathcal{D}) = \mathbb{E}_{(\boldsymbol{x}, \boldsymbol{y}) \sim \mathcal{D}} \left[ r(\boldsymbol{x}, \boldsymbol{y}; \theta) + \log \pi_{\text{ref}}(\boldsymbol{y}|\boldsymbol{x}) \right] - \mathbb{E}_{\boldsymbol{x} \sim \mu(\cdot), \boldsymbol{y} \sim \pi_{r_\theta}^*(\cdot|\boldsymbol{x})} \left[ r(\boldsymbol{x}, \boldsymbol{y}; \theta) - D_{\text{KL}}\left(\pi_{r_\theta}^*(\cdot|\boldsymbol{x}) \| \pi_{\text{ref}}(\cdot|\boldsymbol{x})\right) \right].$$

Then we have the following result:

$$|L(\theta) - \widehat{L}(\theta; \mathcal{D})| = \left| \mathbb{E}_{\boldsymbol{x} \sim \mu(\cdot), \boldsymbol{y} \sim \pi^{\text{E}}(\cdot|\boldsymbol{x})} \left[ r(\boldsymbol{x}, \boldsymbol{y}; \theta) + \log \pi_{\text{ref}}(\boldsymbol{y}|\boldsymbol{x}) \right] - \mathbb{E}_{(\boldsymbol{x}, \boldsymbol{y}) \sim \mathcal{D}} \left[ r(\boldsymbol{x}, \boldsymbol{y}; \theta) + \log \pi_{\text{ref}}(\boldsymbol{y}|\boldsymbol{x}) \right] \right|.$$

According to Assumption 4.1, we obtain that $0 \leq r(\boldsymbol{x}, \boldsymbol{y}; \theta) \leq C_r$ and $C_p \leq \log \pi_{\text{ref}}(\boldsymbol{y}|\boldsymbol{x}) < 0$. Then by applying Hoeffding's inequality, for any $\epsilon > 0$, we have the following result:

$$P\left( \left| \mathbb{E}_{\boldsymbol{x} \sim \mu(\cdot), \boldsymbol{y} \sim \pi^{\text{E}}(\cdot|\boldsymbol{x})} \left[ r(\boldsymbol{x}, \boldsymbol{y}; \theta) + \log \pi_{\text{ref}}(\boldsymbol{y}|\boldsymbol{x}) \right] - \mathbb{E}_{(\boldsymbol{x}, \boldsymbol{y}) \sim \mathcal{D}} \left[ r(\boldsymbol{x}, \boldsymbol{y}; \theta) + \log \pi_{\text{ref}}(\boldsymbol{y}|\boldsymbol{x}) \right] \right| \geq \epsilon \right) \leq 2 \exp\left( -\frac{2|\mathcal{D}|\epsilon^2}{(C_r - C_p)^2} \right).$$

Then by setting $\delta = 2 \exp\left( -\frac{2|\mathcal{D}|\epsilon^2}{(C_r - C_p)^2} \right)$, with probability greater than $1 - \delta$, we have

$$\left| \mathbb{E}_{\boldsymbol{x} \sim \mu(\cdot), \boldsymbol{y} \sim \pi^{\text{E}}(\cdot|\boldsymbol{x})} \left[ r(\boldsymbol{x}, \boldsymbol{y}; \theta) + \log \pi_{\text{ref}}(\boldsymbol{y}|\boldsymbol{x}) \right] - \mathbb{E}_{(\boldsymbol{x}, \boldsymbol{y}) \sim \mathcal{D}} \left[ r(\boldsymbol{x}, \boldsymbol{y}; \theta) + \log \pi_{\text{ref}}(\boldsymbol{y}|\boldsymbol{x}) \right] \right| \leq (C_r - C_p) \sqrt{\frac{\ln(2/\delta)}{2|\mathcal{D}|}}, \tag{13}$$

where $C_r$ and $C_p$ is the constant defined in Assumption 4.1. According to Eq. (13), we obtain the concentration bound to quantify the approximation between $L(\theta)$ and $\widehat{L}(\theta; \mathcal{D})$ as below:

$$|L(\theta) - \widehat{L}(\theta; \mathcal{D})| \leq (C_r - C_p) \sqrt{\frac{\ln(2/\delta)}{2|\mathcal{D}|}}, \quad \text{with probability greater than } 1 - \delta.$$

This completes the proof of this lemma. $\qquad\square$

## C    PROOF OF LEMMA 5.1

*Proof.* In the surrogate estimation problem $\widehat{L}(\theta; \mathcal{D})$ defined in Eq. (6), the policy $\pi_{r_\theta}^*$ corresponds to the solution of the policy optimization problem Eq. (5b). One can show that $\pi_{r_\theta}^*$ is the following closed-form expression:

$$\pi_{r_\theta}^*(\boldsymbol{y}|\boldsymbol{x}) = \frac{\pi_{\text{ref}}(\boldsymbol{y}|\boldsymbol{x}) \exp\left(r(\boldsymbol{x}, \boldsymbol{y}; \theta)\right)}{\sum_{\tilde{\boldsymbol{y}}} \pi_{\text{ref}}(\tilde{\boldsymbol{y}}|\boldsymbol{x}) \exp\left(r(\boldsymbol{x}, \tilde{\boldsymbol{y}}; \theta)\right)}, \quad \forall \boldsymbol{x}, \boldsymbol{y}. \tag{14}$$

Plugging Eq. (14) into Eq. (6), we obtain:

$$\max_\theta \widehat{L}(\theta; \mathcal{D}) = \mathbb{E}_{(\boldsymbol{x}, \boldsymbol{y}) \sim \mathcal{D}} \left[r(\boldsymbol{x}, \boldsymbol{y}; \theta) + \log \pi_{\text{ref}}(\boldsymbol{y}|\boldsymbol{x})\right] - \mathbb{E}_{\boldsymbol{x} \sim \mu} \left[\log\left(\sum_{\tilde{\boldsymbol{y}}} \pi_{\text{ref}}(\tilde{\boldsymbol{y}}|\boldsymbol{x}) \exp\left(r(\boldsymbol{x}, \tilde{\boldsymbol{y}}; \theta)\right)\right)\right] \tag{15}$$

Calculating the derivative we get

$$\max_\theta \widehat{L}(\theta; \mathcal{D}) = \mathbb{E}_{(\boldsymbol{x}, \boldsymbol{y}) \sim \mathcal{D}}[\nabla_\theta r(x, y; \theta)] - \mathbb{E}_{\boldsymbol{x} \sim \mu} \left[\nabla_\theta \log\left(\sum_{\tilde{\boldsymbol{y}}} \pi_{\text{ref}}(\tilde{\boldsymbol{y}}|\boldsymbol{x}) \exp\left(r(\boldsymbol{x}, \tilde{\boldsymbol{y}}; \theta)\right)\right)\right]$$

$$= \mathbb{E}_{(\boldsymbol{x}, \boldsymbol{y}) \sim \mathcal{D}}[\nabla_\theta r(x, y; \theta)] - \mathbb{E}_{\boldsymbol{x} \sim \mu} \left[\sum_{\boldsymbol{y}} \frac{\pi_{\text{ref}}(\boldsymbol{y}|\boldsymbol{x}) \exp\left(r(\boldsymbol{x}, \boldsymbol{y}; \theta)\right)}{\sum_{\tilde{\boldsymbol{y}}} \pi_{\text{ref}}(\tilde{\boldsymbol{y}}|\boldsymbol{x}) \exp\left(r(\boldsymbol{x}, \tilde{\boldsymbol{y}}; \theta)\right)} \nabla_\theta r(\boldsymbol{x}, \boldsymbol{y}; \theta)\right]$$

$$= \mathbb{E}_{(\boldsymbol{x}, \boldsymbol{y}) \sim \mathcal{D}}[\nabla_\theta r(\boldsymbol{x}, \boldsymbol{y}; \theta)] - \mathbb{E}_{\boldsymbol{x} \sim \mu, \boldsymbol{y} \sim \pi_\theta^*(\cdot|\boldsymbol{x})}[\nabla_\theta r(\boldsymbol{x}, \boldsymbol{y}; \theta)].$$

The proof is completed. □

