# OpenReview forum: "From Demonstrations to Rewards:  Alignment Without Explicit Human Preferences"
_ICLR.cc/2025/Conference — Submitted to ICLR 2025_

### Official Review · Reviewer_u6FQ · 2024-10-21

**Soundness:** 3
**Presentation:** 2
**Contribution:** 3
**Rating:** 5
**Confidence:** 4

**Summary:**

This paper shows how one my instantiate the inverse reinforcement learning (IRL) framework in the space of large language models, in place of traditional supervised learning and RLHF methods. The authors propose that a key benefit of this method is the ability to learn from implicit information present in demonstration data, and thus does not require explicit human preference labels as is commonly required in RLHF methods. Similar to conventional IRL, the authors phrase the problem as a bi-level optimization problem, where the reward and policy networks are optimized iteratively to mimic expert behavior. The paper's main contributions are a theoretical derivation of IRL in the space of LLMs, as well as empirical results on TL;DR and UltraChat datasets.

**Strengths:**

* The bi-level optimization framework introduced in this paper is a creative application of IRL to settings where it is not typically applied. The proposed framework replaces the need for explicit preference-based reward models without the need for any explicit preference feedback by leveraging IRL. From another perspective, this paper also extends IRL methods to new domains like summarization (TL;DR) and dialogue generation (UltraChat).
* The paper provides sound mathematical formulations and demonstrates theoretical convergence guarantees, in addition to their empirical results. Though the idea is not particularly novel (as such sample complexity bounds have been proved in other avenues of IRL), of interest is the surrogate estimation problem in Eq 6, which the authors use to derive that the IRL for LLM alignment objective (Eq. 5) can be approximated with a finite set of expert data.
* While citing SPIN as a related work, this paper shows how SPIN's approach is a specific case of their broader IRL-based framework. This may be of particular interest for the community to develop new theoretical insights into expanding the theoretical guarantees of IRL in the space of LLMs.
* The empirical experiments include reasonable metrics and are evaluated over multiple datasets, and the authors demonstrate that their IRL method beats baseline models such as SPIN over almost all metrics.
* The experiment implementations are well described, up to actual implementations, codebases, and hyperparameters, lending itself for future reproducibility.
* The paper has a well-written introduction with solid related works. The problem is adequately motivated and contributions are stated clearly.

**Weaknesses:**

* As with typical IRL, there is a dependence on high-quality demonstration data (indeed, this is part of the finite-sample approximation proof using Eq. 6). It would be nice to see more discussion between the costs of acquiring high-quality demonstration data needed for the proposed IRL approach to work, versus the cost of obtaining human preference labels for RLHF methods to work.
* Related to the point above, It would also be nice to see how the proposed method does with varying amounts of expert demonstration data (in comparison to, say, a method like IQLearn).
* While this paper builds atop SPIN and IRL approaches, there feels to be a lack of novel insights being introduced in the algorithmic design.
* It is well known that IRL approaches are typically rather hard to scale, due to needing to solve the hard exploration problem to optimize the reward in the inner loop. It is a bit unclear how the proposed method scales in comparison to prior work. It would have been nice to see some discussion about training/memory costs of running something like PPO in the inner loop of this bi-level optimization.
* While I understand there are slightly different assumptions being made (implicit vs explicit preference labels), it would be nice to see how current RLHF models do in comparison to the metrics produced by SPIN and the proposed method.

**Questions:**

1. Were there experiments investigating how the proposed method handles varying qualities of demonstration data? In typical imitation learning terms, how does the proposed method function under noisy or suboptimal data?
2. How do the synthetic preference data that your method generate compare with actual human preferences?
3. What is the computational cost of your bi-level optimization approach compared to standard RLHF pipelines? Have you profiled the memory and runtime performance of your method? For example, how long does one iteration of SPIN vs one iteration of IRL take?
4. Assumption 4.1 (Line 253) assumes bounded reward scores. Can you explain the intuitive meaning of this assumption and how it holds in practice? Does this assumption limit the applicability of your method to only certain types of reward functions?

---

> ### Author Response · Authors · 2024-11-27
>
> We thank the reviewer for the detailed review of the paper and the valuable feedback below. We hope you will consider revising your score in light of our responses. Please let us know if you have more questions.
>
> > As with typical IRL, there is a dependence on high-quality demonstration data (indeed, this is part of the finite-sample approximation proof using Eq. 6). It would be nice to see more discussion between the costs of acquiring high-quality demonstration data needed for the proposed IRL approach to work, versus the cost of obtaining human preference labels for RLHF methods to work.
> Related to the point above, It would also be nice to see how the proposed method does with varying amounts of expert demonstration data (in comparison to, say, a method like IQLearn).
>
> That is a great comment. We would like to clarify that the proposed method does not collect additional demonstration data. Our method only utilizes SFT data set to train a reward model and further improve the policy over the SFT model. The data collection cost of our proposed method is lower than standard RLHF method, since we only collect demonstrations for the IRL approach (similar to the SFT stage of RLHF). In practice, many public models are trained with more demonstrations compared with the preference data they use. One potential resason is that demonstrations can sometimes be collected from existing datasets or created by experts in a more efficient manner or even synthesized automatically, which can be cost-effectively. In contrast, collecting preference data requires human annotators to evaluate, compare, and rank which can be time-consuming and costly due to the need for careful human judgment. We include several public models and amount of demonstration and preference data they use for alignment:
> | Model Name | Demonstration Data | Preference Data |
> | -------- | ------- | ------- |
> | HuggingFaceH4/zephyr-7b-beta | HuggingFaceH4/ultrachat_200k (200k) | HuggingFaceH4/ultrafeedback (60k) |
> | HuggingFaceTB/SmolLM2-1.7B-Instruct | HuggingFaceTB/smoltalk 1.1M | HuggingFaceH4/ultrafeedback (60k) |
> |HuggingFaceH4/starchat2-15b-v0.1| HuggingFaceH4/airoboros-3.2 (58k) + HuggingFaceH4/Code-Feedback (65k) + HuggingFaceH4/orca-math-word-problems-200k (200k) + HuggingFaceH4/SystemChat (6.5k) + HuggingFaceH4/capybara (16k)  | HuggingFaceH4/ultrafeedback_binarized (60k) + HuggingFaceH4/orca_dpo_pairs (12.4k) |
>
> From the above table, we indeed observe that typically the size of the demonstration data is larger than the preference data, therefore the  data 'unbalanced' situation often arises.
>
> For our two experiment setting in Sec. 5, we include the numbers of demonstration and preference data in the following table.
> | Experiment setting | Demonstration Data | Preference Data |
> | -------- | ------- | ------- |
> | 1b/2.8b | 10k | 5k/10k |
> | 7b | 208k | 61.1k |
>
> > While this paper builds atop SPIN and IRL approaches, there feels to be a lack of novel insights being introduced in the algorithmic design.
>
> In this paper, r, we aim to design one easy-to-use algorithmic pipeline which can leverage demonstrations and can be compatible with the current RLHF pipeline. Hence, the focus of this paper is not to design a new IRL algorithm but make IRL easy to use and compatible with the RLHF pipeline.
>
> > It is well known that IRL approaches are typically rather hard to scale, due to needing to solve the hard exploration problem to optimize the reward in the inner loop. It is a bit unclear how the proposed method scales in comparison to prior work. It would have been nice to see some discussion about training/memory costs of running something like PPO in the inner loop of this bi-level optimization.
>
> We would like to point out that in the inner loop, we can adopt the state-of-the-art policy optimization algorithms. When the model size is small, training PPO can be affordable as we show in Sec. 6.1 where we run PPO to finetune 1B models. In Sec. 6.2, when the model size is 7B where PPO is not affordable for our device (8 * A100 (80GB)), we use Best-of-N as one computationally efficient algorithm to replace PPO in the inner loop.
>
> > While I understand there are slightly different assumptions being made (implicit vs explicit preference labels), it would be nice to see how current RLHF models do in comparison to the metrics produced by SPIN and the proposed method.
>
> In our paper, in order to compare SPIN with our proposed method, we use two metrics: 1) reward performance measured by RewardBench, 2) policy performance measured by Open LLM Leaderboard. We have reported the numerical results in Tab. 1 and Tab. 2 of our paper.

---

> ### Author Response · Authors · 2024-11-27
>
> > Were there experiments investigating how the proposed method handles varying qualities of demonstration data? In typical imitation learning terms, how does the proposed method function under noisy or suboptimal data?
>
> There are some differences between LLM alignment and the typical setting of imitation learning.  Notably, in the typical setting of imitation learning, usually one expert demonstration can train one well-estimated reward model so that its corresponding optimal policy can achieve expert-level performance. However, in LLM alignment, considering that the tasks is more complicated and the model has much more parameters compared with the control tasks in standard imitation learning, it is important to use sufficient demonstrations to train the reward / policy. Compared with typical imitation learning terms, this paper designs a framework to leverage demonstrations to train a good reward model for LLM alignment. Since our proposed approach regards the demonstration data as preferred data while noisy or suboptimal data can inject noises into the synthetic preference data, we did not test the performance of our proposed approach under noisy or suboptimal data.
>
> > How do the synthetic preference data that your method generate compare with actual human preferences?
>
> To show the comparison between the synthetic preference data and the actual human preferences, we would like to point to our experiment of the reward training in Sec. 6.1. In the TL;DR experiment, the reward model we train using the synthetic preference data achieves 58.8% accuracy in TL;DR validation dataset. In [2], the author has trained several 1B reward model using the annotated preference dataset openai/summarize_from_feedback. In their reported evaluation results of their estimated 1B reward model in [Tracked Wandb Logs](https://wandb.ai/costa-huang/tldr_summarize/runs/z6v2q8nx?nw=nwusercostahuang), the accuracy over the validation dataset is 62.84%, which is close to our IRL reward model. Considering the training of our IRL reward model does not involve the usage of the preference dataset, the reported accuracy shows that IRL indeed learns a good reward model from the synthetic preference data.
>
> > What is the computational cost of your bi-level optimization approach compared to standard RLHF pipelines? Have you profiled the memory and runtime performance of your method? For example, how long does one iteration of SPIN vs one iteration of IRL take?
>
> Our proposed pipeline shares similar memory cost with standard RLHF pipelines since our proposed method follows similar reward update scheme and policy update scheme with standard RLHF pipelines. For the training time, one iteration of SPIN takes 8 hours for optimizing 7B models with UltraChat and one iteration of IRL takes 10 hours since we uses Best-of-N as one computational efficient policy optimization method in our pipeline for training 7B models.
>
> > Assumption 4.1 (Line 253) assumes bounded reward scores. Can you explain the intuitive meaning of this assumption and how it holds in practice? Does this assumption limit the applicability of your method to only certain types of reward functions?
>
> In practical applications, the range of the reward output will be controlled to keep the PPO update stable. As discussed in [1] (see Detail 15 in Sec. 6), the output of reward model will be normalized before feeding into PPO training.
>
> [1] Huang, Shengyi, et al. "The N+ Implementation Details of RLHF with PPO: A Case Study on TL; DR Summarization." arXiv preprint arXiv:2403.17031 (2024).

---

> > ### Comment · Reviewer_u6FQ · 2024-12-02
> >
> > I thank the authors for their responses to my questions. I have gotten a better understanding of where this paper is placed relevant to related work, and I highly recommend the authors add their clarifications (especially to those regarding the quality and quantity of demonstration data) to the revised manuscript. I still feel that the contributions are incremental, and will thus maintain my original score.

---

### Official Review · Reviewer_AVi1 · 2024-10-29

**Soundness:** 3
**Presentation:** 3
**Contribution:** 2
**Rating:** 5
**Confidence:** 4

**Summary:**

This paper presents an approach to aligning language models with human preferences using only demonstration data, bypassing the need for explicit preference data typically required in RLHF. Drawing on Inverse Reinforcement Learning, the method iteratively learns both a reward and policy model solely from demonstration data, treating ground-truth responses as implicit preferences. This setup reduces the data complexity associated with traditional RLHF pipelines, which rely on human-labeled preference data for reward modeling.

**Strengths:**

- The paper is generally well-written and clear.
- The experimental setup is described in detail, providing clarity on the methodology.
- The approach achieves model alignment using only demonstration data, which could simplify data requirements compared to traditional RLHF that demands extensive preference labeling

**Weaknesses:**

Major:
- The proposed approach appears to be a specific case of iterative DPO. In iterative DPO, two responses are generated using the current model, and an expert LLM provides the preference judgment, creating the preference dataset needed for optimization. The proposed approach differs only in:
    * treating the ground truth in the demonstration data as the preferable response
    * using PPO rather than DPO
- To establish the contribution's originality, the authors should comprehensively describe and compare their method against iterative DPO. Specifically,
    * Evaluate the performance of iterative DPO without using any demonstration data on the same datasets.
    * Evaluate an approach that uses ground-truth data as preferred responses but leverages DPO for preference optimization, to isolate the impact of using PPO versus DPO within the proposed setup.

Minor:
- The reward accuracy shown in Figure 2(a) appears notably low. A random reward model would be expected to achieve around 50% accuracy by chance, yet the highest accuracy achieved here is only 58.8%.
- The performance drop from 79.19% to 48.04% on the "Chat" task after one SPIN iteration seems unexpected. This drop suggests potential issues with fine-tuning stability or model configuration. Could author provide some explanation about that?

**Questions:**

- What is the performance of iterative DPO without using any demonstration data on the two datasets?
- What is the difference in performance of using PPO versus DPO in optimization?
- Some other questions are mentioned in weakness.

---

> ### Author Response · Authors · 2024-11-27
>
> We thank the reviewer for the detailed review of the paper and the valuable feedback. We hope you will consider revising your score in light of our responses. Please let us know if you have more questions.
>
> ****
>
> > * The proposed approach appears to be a specific case of iterative DPO. In iterative DPO, two responses are generated using the current model, and an expert LLM provides the preference judgment, creating the preference dataset needed for optimization. The proposed approach differs only in:
> > * treating the ground truth in the demonstration data as the preferable response
> using PPO rather than DPO
> > * using PPO rather than DPO
>
> We would like to note that our proposed IRL method is **not** a specific case of iterative DPO. Thereare a few key differences, listed below:
>
> - The proposed algorithm involves explicit reward modeling, and reward updates, while iterative DPO does not have reward modeling and updates;
> - The proposed algorithm only relies on demonstration data and self generated data, while not replying on any external reward models or judge models, while iterative DPO related works has to have those external judges to perform the annotation, and they typically do not assume the access to demonstration/SFT data.
>
>
>
> To ellaborate the first point above, the key point of designing inverse RL based algorithm is to model and train explicit reward models *from the demonstration dataset*, while learning the policy at the same time. This is not the case for iterative DPO-based algorithm because 1) DPO algorithm itself requires preference data, not demonstration data 2) these algorithms do not explicitly learn reward models.
>
> Let us then ellaborate the second point above. According to the reviewer's comments, iterative DPO relies on "an expert LLM provides the preference judgment, creating the preference dataset needed for optimization". However, in our proposed method, there is no "expert LLM as judgement". The focus of this paper is to design an efficient algorithm to train a reward model and a policy through learning from demonstration data in LLM alignment. Essentially we are building our own 'judge' (i.e., the reward model) and then train the policy. This is very different as compared to the setting mentioned by the reviewer.
>
> Moreover, to provide a more direct comparison between iterative DPO and our proposed method under the same setting, lett's assume there is no expert LLM to provide preference judgments, but instead we always treat the demonstration data as the preferable response in iterative DPO, while treating the generations from the current model as the non-preferable response to construct synthetic preference. Note this is different than the typical iterative DPO setting since typically there is no demonstration data assumed in the iterative DPO; but since here we are considering using demonstration data only, this is reasonable.   Then running the iterative DPO algorithm on this dataset becomes **equivalent** to running Self-Play fIne-tuNing (SPIN) [1], which is not IRL. The key difference here is that in our proposed algorithm based on IRL, explicit reward model is constructed and updated iteratively.  We have performaed extensive experiments to compare our proposed algorithm with SPIN, and demonstrated its advantage. Therefore we can conclude that the proposed approach is not a specific case of iterative DPO.
>
> [1] Chen, Zixiang, et al. "Self-Play Fine-Tuning Converts Weak Language Models to Strong Language Models." *Forty-first International Conference on Machine Learning.*
>
> > To establish the contribution's originality, the authors should comprehensively describe and compare their method against iterative DPO. Specifically,
> > * Evaluate the performance of iterative DPO without using any demonstration data on the same datasets.
> > * Evaluate an approach that uses ground-truth data as preferred responses but leverages DPO for preference optimization, to isolate the impact of using PPO versus DPO within the proposed setup.
>
> As we mentioned in our response to the reviewer's previous comment,, when there is no expert LLM as judgement and the demonstration is treated as the prefered response, then iterative DPO turns to be the algorithm SPIN. In the experiment set for training 7B Mistral models using the UltraChat dataset, we did extensive experiment to compare the proposed approach with SPIN. We hope our response can address the reviewer's concern.

---

> ### Author Response · Authors · 2024-11-27
>
> > The reward accuracy shown in Figure 2(a) appears notably low. A random reward model would be expected to achieve around 50% accuracy by chance, yet the highest accuracy achieved here is only 58.8%.
>
> We would like to justify that 58.8% accuracy in TL;DR is ok for a 1B reward model, especially there is no ground-truth preference dataset available. In [2], the author has trained several 1B reward model using the annotated preference dataset openai/summarize_from_feedback. In their reported evaluation results of their estimated 1B reward model in [Tracked Wandb Logs](https://wandb.ai/costa-huang/tldr_summarize/runs/z6v2q8nx?nw=nwusercostahuang), the accuracy over the validation dataset is 62.84%, which is close to our IRL reward model. Considering the training of our IRL reward model does not involve the usage of the preference dataset, the reported accuracy shows that IRL indeed learns a good reward model from demonstration data. We are not aware of any othe reward model only uses demonstration dataset in this case.
>
>
> [2] Huang, Shengyi, et al. "The N+ Implementation Details of RLHF with PPO: A Case Study on TL; DR Summarization." arXiv preprint arXiv:2403.17031 (2024).
>
>
> > The performance drop from 79.19% to 48.04% on the "Chat" task after one SPIN iteration seems unexpected. This drop suggests potential issues with fine-tuning stability or model configuration. Could author provide some explanation about that?
>
> SPIN is a variant of iterative DPO which does not train an explicit reward model. We use the SPIN's implicit reward model $r(x, y) := \log \pi(y|x) - \log \pi_{\text{ref}}(y|x)$ to evaluate the performance in RewardBench. One reason why SPIN does not obtain an accurate implicit reward model is due to the fact that the implicit reward model induced by DPO has limited generalization capability and can easily overfit preference data [3]. This is exactly the reason why we consider training one explicit reward model in our IRL pipeline.
>
> [3] Lin, Yong, et al. "On the limited generalization capability of the implicit reward model induced by direct preference optimization." arXiv preprint arXiv:2409.03650 (2024).
>
> ****
> > What is the performance of iterative DPO without using any demonstration data on the two datasets?
>
> In our setting, there is no expert LLM as judegement or available human annotations. Hence, the only way to run iterative DPO is follow the SPIN algorithm. We have shown the results in Tab. 2.
>
> > What is the difference in performance of using PPO versus DPO in optimization?
>
> In Tab.2, we can check the comparison between IRL Iter1 and SPIN Iter1 as PPO versus DPO. This is due to the fact that IRL Iter1 and SPIN Iter1 use the same synthetic preference data while IRL Iter1 first trains a reward model and then uses policy optimization methods to finetune the policy while SPIN Iter1 directly runs DPO using the synthetic preference data (demonstration vs SFT-Model-Generations).

---

> > ### Comment · Reviewer_AVi1 · 2024-11-28
> >
> > Thank you for the response. The authors' clarification aligns with my original understanding of the paper. However, using DPO or a reward model combined with PPO should not be regarded as a primary contribution. While the use of demonstration data for alignment is interesting, it does not demonstrate sufficient novelty. Additionally, I find the comparison in Table 1 problematic, as it might mislead the readers by comparing the performance of implicit rewards with a standard reward model, which is not a fair comparison. Therefore, I will maintain my score.

---

### Official Review · Reviewer_HKyJ · 2024-10-31

**Soundness:** 3
**Presentation:** 3
**Contribution:** 3
**Rating:** 6
**Confidence:** 4

**Summary:**

The authors apply existing Inverse RL (IRL) methods to the LLM fine-tuning problem, yielding a method of improving LLM alignment with only demonstration data that empirically outperforms SFT and SPIN.
Thus, the old result of IRL beating behavioral cloning is replicated in the LLM setting.

Theoretical connections between their method and SPIN are made.

**Strengths:**

Paper nicely applies existing IRL work to the LLM fine-tuning problem, yielding new insights into both.

A theoretical connection between the new method and existing work is made.

Empirical evaluation of the method is strong, it outperforms existing baselines.

The related work and preliminaries section nicely explains previous relevant methods and results, and why one might expect the proposed method to work well a priori.

Method and its motivation is clearly explained.

Strong empirical results to support the claim this method outperforms SFT/SPIN.

**Weaknesses:**

An analogous equation to 6 is derived in the maximum-entropy IRL literature, so it seems reasonable to be optimising it in the downstream algorithm.
However, the derivation of equation 6 from equation 5a is not clear, and lemma 4.1 is incredibly weak---it seems $C_p$ could be extremely negative and $C_r$ very positive, making the bound rather vacuous.
Furthermore, since equation 5/5a is not later referred to in the paper after lemma 4.1, it's not clear what point is being made by formulating the problem this way, beyond the sense that something approximating MLE is probably a good thing.
It seems as though you may as well just start with equation 6, cite the relevant literature as inspiration, and move on.

Equation 6 is not actually used for the optimisation process, equation 11 is.
And like going from 5a to 6, the derivation of why optimising for equation 11 is equivalent to optimising that of 6 is not given mathematically, instead motivated by intuitive reasoning.
They are in fact **not** equivalent.
The log-sigmoid in equation 11 will decrease the loss of policy-trajectory / demonstration pairs that are already well differentiated in reward, whereas equation 6 will always be trying to push them apart with 'equal force'.
Thus, whilst they would provide the same direction of update for any given pair, they would not give the same magnitude.
When this effect is considered over a dataset of many demonstrations and policy-trajectories, the overall path taken in reward model parameter space by the optimisation procedure will differ as it moves different amounts in each of the different directions.
Whilst this is unlikely to significantly harm the method---and is probably actually a desirable property---it further breaks the connection between what's actually been done (optimising equation 11) and what has been claimed to have been done (optimising equation 5a).

Numerical results, both used in tables and graphs, do not have error bars, nor is it clear that more than a single random seed has been used.

Whilst the results in the paper support the claim that the proposed method is better than SFT/SPIN, since SFT is usually supplemented by RLHF, it would be interesting to see a comparison of the IRL method + RLHF vs SFT + RLHF, or even the IRL method vs SFT + RLHF.
It would be disappointing if the advantages of this method over SFT were rendered obsolete by downstream RLHF tuning, and exciting if this method could achieve alignment comparable to SFT + RLHF without needing any preference data itself.

## Errata
* Line 147/148, "...there are **wo** prominent classes..." -> "...there are **two** prominent classes..."

**Questions:**

Please either clarify the claims made with relation to what the method actually implements, or provide more detail on the derivation from equation 5a to 6 to 11, and what has changed / been lost along the way.

---

> ### Author Response · Authors · 2024-11-27
>
> We thank the reviewer for the detailed review of the paper and the valuable feedback. Below, we address the reviewer's questions in a point-by-point manner.
>
> ****
>
> > An analogous equation to 6 is derived in the maximum-entropy IRL literature, so it seems reasonable to be optimising it in the downstream algorithm. However, the derivation of equation 6 from equation 5a is not clear, and lemma 4.1 is incredibly weak---it seems $C_p$ could be extremely negative and $C_r$ very positive, making the bound rather vacuous. Furthermore, since equation 5/5a is not later referred to in the paper after lemma 4.1, it's not clear what point is being made by formulating the problem this way, beyond the sense that something approximating MLE is probably a good thing. It seems as though you may as well just start with equation 6, cite the relevant literature as inspiration, and move on.
>
> The derivation from equation 5 to equation 6 could be find in Appendix. The formulation in equation 5/5a models the reward learning / policy finetuning problem from a maximum likelihood perspective and equation 6 is its fineite-sample surrogate problem. In lemma 4.1, we are trying to provide one asymptotic bound to show that when there is sufficient demonstration data, the surrogate problem can be an accurate approximation to the MLE problem.
>
> > Equation 6 is not actually used for the optimisation process, equation 11 is. And like going from 5a to 6, the derivation of why optimising for equation 11 is equivalent to optimising that of 6 is not given mathematically, instead motivated by intuitive reasoning. They are in fact not equivalent. The log-sigmoid in equation 11 will decrease the loss of policy-trajectory / demonstration pairs that are already well differentiated in reward, whereas equation 6 will always be trying to push them apart with 'equal force'. Thus, whilst they would provide the same direction of update for any given pair, they would not give the same magnitude. When this effect is considered over a dataset of many demonstrations and policy-trajectories, the overall path taken in reward model parameter space by the optimisation procedure will differ as it moves different amounts in each of the different directions. Whilst this is unlikely to significantly harm the method---and is probably actually a desirable property---it further breaks the connection between what's actually been done (optimising equation 11) and what has been claimed to have been done (optimising equation 5a).
>
> That is a correct observation.We acknowledge that the connection between eq(10) and eq(11) is not mathematically. The reason why we want to show the connection is to show that the proposed IRL method can be compatiable with the standard RLHF pipeline. We will update the experiment the show the difference between training reward model using eq(10) and eq(11).
>
> > Numerical results, both used in tables and graphs, do not have error bars, nor is it clear that more than a single random seed has been used.
>
> Thanks for the suggestion. To address this comment, we will include the error bars in the final version.
>
> > 4. Whilst the results in the paper support the claim that the proposed method is better than SFT/SPIN, since SFT is usually supplemented by RLHF, it would be interesting to see a comparison of the IRL method + RLHF vs SFT + RLHF, or even the IRL method vs SFT + RLHF. It would be disappointing if the advantages of this method over SFT were rendered obsolete by downstream RLHF tuning, and exciting if this method could achieve alignment comparable to SFT + RLHF without needing any preference data itself.
>
> In our proposed IRL method, we leverage the demonstration data for reward training. Indeed, RLHF and IRL can be compatible which is the reason why we use the reward update rule eq. (11) in our paper. We have show the numerical results for comparing the reward accrucy between RLHF + RLHF vs SFT + RLHF:
> | Reward Model | Chat | Chat Hard | Safety | Reasoning | Average |
> | -------- | ------- | ------- | ------- | ------- | ------- |
> | SFT + RLHF | 95.11% | 56.58% | 63.69% | 69.22% | 71.15% |
> | RLHF + IRL | 94.41% | 55.37% | 63.98% | 76.75% | 72.63% |

---

> > ### Comment · Reviewer_HKyJ · 2024-11-30
> >
> > Thank you for your response. I am disappointed to see that an updated draft at least correcting typos pointed out by myself and other reviewers has not been provided.
> >
> > "The derivation from equation 5 to equation 6 could be find in Appendix."
> > Where exactly?
> > All I see is sections on "Experiment Details", "Proof of Lemma 4.1", and "Proof of Lemma 5.1".
> > None of these seem to contain said derivation.
> >
> > "The formulation in equation 5/5a models the reward learning / policy finetuning problem from a maximum likelihood perspective and equation 6 is its fineite-sample surrogate problem. In lemma 4.1, we are trying to provide one asymptotic bound to show that when there is sufficient demonstration data, the surrogate problem can be an accurate approximation to the MLE problem."
> > I see, thank you for the clarification.
> > The bound still does not seem particularly useful, the logprob of a completion is probably of order ~-100, its reward of order ~1, and a reasonable value of $\delta$ is perhaps 0.05.
> > To bound the likelihood difference to within $\pm 0.1$ you'd need ~2 million training examples.
> >
> > "That is a correct observation.We acknowledge that the connection between eq(10) and eq(11) is not mathematically. The reason why we want to show the connection is to show that the proposed IRL method can be compatiable with the standard RLHF pipeline. We will update the experiment the show the difference between training reward model using eq(10) and eq(11)."
> > My concern still stands and I note the paper has not been updated.
> >
> > "Thanks for the suggestion. To address this comment, we will include the error bars in the final version."
> > This has not been done.
> > Did you use multiple random seeds for the experiments listed in the paper or did you not?
> > If so how many?
> >
> > "In our proposed IRL method, we leverage the demonstration data for reward training. Indeed, RLHF and IRL can be compatible which is the reason why we use the reward update rule eq. (11) in our paper. We have show the numerical results for comparing the reward accrucy between RLHF + RLHF vs SFT + RLHF:"
> > Thank you for these results.
> > It's interesting that some metrics increase and some decrease vs IRL alone.
> > However it does seem the overall advantage of IRL vs SFT is somewhat limited / reduced if RLHF is also being applied.
> >
> > Whilst you have have addressed some concerns, many remain.

---

### Official Review · Reviewer_8rft · 2024-11-03

**Soundness:** 2
**Presentation:** 2
**Contribution:** 2
**Rating:** 3
**Confidence:** 4

**Summary:**

This paper proposes to learn an LLM reward model from demonstrations data only, rather than preference data as is traditional in RLHF finetuning procedures. The authors show that such a reward estimator can be 1) learned effectively and 2) used to predict preferences well in a held out dataset when compared to baselines like SFT and SPIN.

**Strengths:**

The paper has the following strengths:

1. Alignment to human preferences, learned from human data, is a question that is very timely in the community. The paper does a good job of overviewing the current state of approaches used in the community, including if/when they fall short of theoretically desired behaviors. However, it would be nice if the explicitly link between the generative IRL methods proposed in Sec. 2 were explicitly tied to the author's formulation in greater detail.
2. The proposed formulation is pretty general and therefore easy to intuitively understand - the proposed reward estimator should learn a reward function that produces a policy that best serves as a MLE over the demonstrations dataset.

**Weaknesses:**

I have a number of concerns, which I will summarize into major and minor:

Major Weaknesses:
1. While theoretically sound, the proposed approach is intractable in most situations due to the bi-level optimization problem of both estimating the reward as well as the policy in each update (reflecting similar challenges with deploying meta-learning techniques like MaML, which follow a similar bi-level optimization procedure). As a result, the proposed algorithm for actually optimizing this objective makes a number of assumptions which I think are not likely to hold in practice, such as limited batch sampling for updating the reward estimator and policy parameters.
2. Evaluation is not as thorough as one may hope for a LLM paper (the authors only studied two datasets with one pretrained model checkpoint). For a contribution that is mostly algorithmic, one would expect a more diverse set of tasks (say, HHH or Coding) as well as multiple models checkpointed and their baselines evaluated.

Minor Weaknesses:
1. There are various typos scattered throughout the paper (e.g. line 148 "wo"->"two" or line 290 "detailw"->"detail"). Please ensure the resulting revisions are spell-checked!
2. Please ensure the authors are citing/discussing meta-learning techniques and their shortcomings in related work (as it's particularly important for understanding the limits of bi-level policy and reward optimization procedures such as the one the authors are proposing).

**Questions:**

1. Can the authors please evaluate their proposed approach more thoroughly, or if not, explain why they only evaluated the proposed algorithm on two tasks with one set of checkpointed parameters?

2. Akin to meta-learning procedures, where the bi-level optimization problem is often intractable in practice, the proposed approach requires a series of simplifications and sampling assumptions to ensure the optimization converges in practice. Can the authors justify which situations would these assumptions cause issues, and when they would expect this to be justifiable when actually training large language models?

---

> ### Author Response · Authors · 2024-11-27
>
> We thank the reviewer for the detailed review of the paper and the feedback.   We hope you'll consider increasing your score after reading our responses as we believe we have addressed all your comments. Please let us know if you have more comments.
>
> ****
>
> > While theoretically sound, the proposed approach is intractable in most situations due to the bi-level optimization problem of both estimating the reward as well as the policy in each update (reflecting similar challenges with deploying meta-learning techniques like MaML, which follow a similar bi-level optimization procedure).
>
> We thank the reviewer for the comment. However, we respectfully disagree with the reviwer's comment. First, we would like to clarify that **it is not true that the proposed method will be computationally intractable due to the bi-level structure**. In contrast, one of the key contribution for this paper is indeed that we have designed a computationally efficient algorithm to align LLM with demonstrations. As a remark, the algorithmic pipeline of our proposed method is as easy as the standard RLHF pipeline which uses preference data instead. In our paper, we have shown detailed update rules for updating reward parameter in eq. (10) and  updating policy parameter in eq. (9). As we discussed in our paper, the proposed reward update rule in (10) and (11) shares similar update scheme as optimizing common reward objective function (i.e. Bradley-Terry model) using preference in eq. (2). Moreover, to implement the policy optimization step in eq. (9), one can utilize any existing policy
> optimization pipeline such as PPO and Best-of-N methods (as we have discussed and implemented in our paper). Hence, implementing our proposed method, where we alternate between reward updates and policy updates, is as easy and straightforward as the standard RLHF pipeline.
>
> >  As a result, the proposed algorithm for actually optimizing this objective makes a number of assumptions which I think are not likely to hold in practice, such as limited batch sampling for updating the reward estimator and policy parameters.
>
> We would like to clarify that the **only** assumption underlying our method is Assumption 4.1, which states that the output of the reward model and the log probability of the reference policy are bounded ( which can be easily applied). Such assumption holds in most scenarios of LLM alignment. For example, [1] also suppose this assumption (see assumption 1).
>
> [1] Ji, Xiang, et al. "Self-Play with Adversarial Critic: Provable and Scalable Offline Alignment for Language Models." arXiv preprint arXiv:2406.04274 (2024).
>
> > Evaluation is not as thorough as one may hope for a LLM paper (the authors only studied two datasets with one pretrained model checkpoint). For a contribution that is mostly algorithmic, one would expect a more diverse set of tasks (say, HHH or Coding) as well as multiple models checkpointed and their baselines evaluated.
>
> In our numerical results, we have two experiment sets: 1) training 1B Pythia model for TL;DR tasks, 2) fine-tuning 7B Mistral model with the UltraChat dataset. Compared with several recent papers [1,2] which study aligning LLM with demonstrations / preference through training 7B Mistral models with UltraChat / UltraFeedback datasets, our paper has presented more numerical experiment sets.
>
> [1] Chen, Zixiang, et al. "Self-Play Fine-Tuning Converts Weak Language Models to Strong Language Models." *Forty-first International Conference on Machine Learning.*
>
> [2] Ji, Xiang, et al. "Self-Play with Adversarial Critic: Provable and Scalable Offline Alignment for Language Models." arXiv preprint arXiv:2406.04274 (2024).
>
> ****
>
> > There are various typos scattered throughout the paper (e.g. line 148 "wo"->"two" or line 290 "detailw"->"detail"). Please ensure the resulting revisions are spell-checked!
>
> We appreciate the reviewer's detailed review. We will correct the typos in our paper.
>
> > Please ensure the authors are citing/discussing meta-learning techniques and their shortcomings in related work (as it's particularly important for understanding the limits of bi-level policy and reward optimization procedures such as the one the authors are proposing).
>
>
> As we clarified in our response to the reviewer's previous comment, our proposed method is computationally efficient and as straightforward as the standard RLHF pipeline. Therefore, it does not suffer from the common limitations of bi-level optimization problems, such as those encountered in meta-learning. That said, we will discuss meta-learning techniques and their shortcomings in the related work section and will expand on this in the final version of the paper.

---

> > ### Comment · Reviewer_8rft · 2024-12-03
> > **Response to authors**
> >
> > I thank the authors for taking the time to rebut my original comment. In particular, the first comment regarding the distillation of the reward update into one analogous to the Bradley-Terry model for iterative DPO was helpful. Unfortunately, this actually reduces the novelty of the paper in my mind, for the contribution is actually therefore even more limited than I originally imagined (as pointed out by the other reviewers, this is now less an interesting albeit impossible bi-level optimization problem and more an iterative DPO update).
> >
> > Moreover, I am concerned about how, despite many reviewers pointing out missing proofs and less-than-ideal writing, that these edits were not made in the revision nor were the existence of said-proofs procured.
> >
> > Therefore, I will keep my score as it is.

---

### Meta-Review · Area_Chair_prrc · 2024-12-20

**Metareview:**

The authors apply existing Inverse RL (IRL) methods to the LLM fine-tuning problem, bypassing the need for explicit preference data typically required in RLHF. The authors show that such a reward estimator can be 1) learned effectively and 2) used to predict preferences well in a held out dataset when compared to baselines like SFT and SPIN.

Reviewers appreciated the discussion of theoretical connections to prior work, the timeliness/importance of the problem, the novelty of the proposed method, and the strong empirical results (relative to baselines). Reviewers commended the paper for being easy to understand and for including replicable code + hyperparameters.

Reviewers had concerns with the tractability of the proposed approach (which requires a bi-level optimization problem), the tightness of Lemma 4.1, and the logical flow of some of the equations, suggesting that the practical method may different notably from its theoretical motivation. Reviewers also questioned whether the proposed method is a special case of iterative DPO. Reviewers recommended evaluation on a more diverse set of tasks, the inclusion of multiple error seeds (for error bars), comparisons with SFT + RLHF (or similar), and some discussion of the cost of collecting high-quality demonstration data.

After a robust discussion, reviewers seemed to agree that the paper was more incremental than it originally seemed. I therefore am recommending that the paper be rejected. I would encourage the authors to include the rebuttal discussion in a revised version of the paper.

**Additional Comments On Reviewer Discussion:**

The authors questioned the concern that the bilevel optimization makes their method intractable, and noted that their comparisons are more thorough than some recent work. They noted that some of the theoretical concerns were addressed with proofs in the appendix, but the difference between theory and practice remains.  The discussion period highlighted that the proposed method is similar to iterative DPO (though the authors argued against this interpretation). There were also new concerns about whether the comparison to baselines was fair in Table 1.

---

### Decision · Program_Chairs · 2025-01-22

Reject